# OpenReview forum: "What Makes In-context Learning Effective for Mathematical Reasoning"
_ICML.cc/2025/Conference — ICML 2025 poster_

### Official Review · Reviewer_voNS · 2025-03-10

**Overall Recommendation:** 4

**Summary:**

In this paper, the authors investigate the theoretical explanation of in-context learning (ICL). They prove that the influence of the demonstrations can be estimated by two factors: LLM-oriented semantic similarity and inference stability of demonstrations. Based on it, they propose a LMS3 method, and the experiments on Llama2 and Llama3 validate the effectiveness of LMS3 under both one-shot and few-shot settings, as well as for both mathematical reasoning and commonsense reasoning tasks.

**Claims And Evidence:**

I have reviewed all the theoretical proofs in this paper and find the claims and results both reasonable and correct. Additionally, the authors have conducted experiments across one-shot to four-shot settings, providing convincing evidence of the proposed method's effectiveness.

**Essential References Not Discussed:**

No. The references are cited and discussed sufficiently

**Experimental Designs Or Analyses:**

Yes, the experiments are conducted using two LLMs, four datasets, and four few-shot settings. Therefore, I think the effectiveness is validated.

**Methods And Evaluation Criteria:**

The proposed method is grounded in the theoretical findings, which makes sense and offers a novel and deeper understanding of ICL. Additionally, the experiments are conducted on three mathematical reasoning datasets and one commonsense reasoning dataset. Therefore, this paper provides strong empirical validation.

**Other Comments Or Suggestions:**

In Eq.(23), lambda_1 is misused, as it was previously introduced to represent an eigenvalue
Line 105, “and offers” should be “and offer”
Line 75, “generate code” should be “generate codes”

**Other Strengths And Weaknesses:**

Strengths:
Frist, building on existing analyses of ICL, this work innovatively derives the theoretical relationship between test loss and demonstrations. Since no prior work has reached a similar conclusion, the novelty of this paper is well justified.

Second, this paper explains how the effectiveness of ICL is determined by both the LLM-oriented Semantic Similarity and Inference Stability of Demonstration. I think this conclusion makes sense and it provides insights for practical applications of ICL.

Third, the experiments are thorough, with the authors using two LLMs as backbones and conducting evaluations across 1-shot to 4-shot settings.

Last, the writing is clear, making the paper easy to understand and reproduce.

Weakness:
The generalizability of the theory and methods in this paper could be further discussed (Q1 below). Besides, I still have some questions regarding the theoretical analyses (Q2 below).

**Questions For Authors:**

Q1: In this paper, the author emphasizes the analysis of situations where ICL is effective for mathematical reasoning. I believe the analysis is sound, but the conclusion could also be applied to other tasks, as the author has demonstrated its effectiveness on CommonsenseQA as well. Therefore, I think the author could further discuss the generalizability of this theory and explain whether it could be effective for other tasks.

Q2: In Eq.(6), I am unsure how the pre-trained data D_pre and the demonstration z0 are optimized simultaneously, because in my understanding, D_pre is for pretraining and z0 is for the gradient update in inference phase?

**Relation To Broader Scientific Literature:**

The authors expand the existing understanding for ICL based on a widely-used theoretical setting. Although the simplification of the softmax function in Eq.(3) may deviate from reality, it has been widely adopted in previous work.

**Theoretical Claims:**

Yes, I have checked the correctness of the proofs in Section 3. They are self-contained and reasonable, providing a good perspective for discussing the influence of demonstrations.

---

> ### Author Rebuttal · Authors · 2025-03-30
>
> We sincerely appreciate your recognition of our reasonable, correct, and self-contained theoretical analysis, the novelty and effectiveness of our method, and our convincing and strong empirical validation.
>
> $\bf{Q1}$: The generalizability of the theory and methods in this paper could be further discussed.
>
> $\bf{A1}$:Thanks for your constructive suggestion! In this paper, we are motivated by the observation that, on several math datasets, LLMs may perform worse in one-shot than in zero-shot setting. Therefore, in our experiments, we use these math datasets for evaluation, which can provide direct evidence of our method’s advantage.
>
> As highlighted in Appendix E, our theories and method can also generalize to other datasets and tasks. This is because they are built upon a general setting of transformer attention layer and the relationship between demonstrations and test samples, which is also suitable to other domains and tasks. For instance, in Section 5.6, we conduct experiments on CommonsenseQA dataset, which is a widely used large-scale commonsense benchmark in ICL research[1,2,3]. From Table 4, our LMS3 still achieves the best performance, which confirms its effectiveness and highlight its generalizability on a broader range of datasets/applications.
>
> Following your suggestions, we will supplement the above discussions in the revised version.
>
> [1] Compositional Exemplars for In-context Learning.
>
> [2] In-Context Learning with Iterative Demonstration Selection.
>
> [3] Rethinking the Role of Demonstrations: What Makes In-Context Learning Work?
>
> $\bf{Q2}$: In Eq.(6), I am unsure how the pre-trained data $D_{pre}$ and the demonstration $z_0$ are optimized simultaneously.
>
> $\bf{A2}$: Thanks for your valuable question! We sincerely apologize for the confusion and would like to clarify as follows. In the “Analogy to Linear Optimization” section, we interpret the influence of adding a demonstration $x$ as follows: we start with a linear function $F$, whose parameters are initialized on the pretraining dataset $D_{pre}$. Introducing $x$ is essentially equivalent to adding a training example $z_0$ to further optimize $F$, after which the optimized $F$ is used to reason the test sample $x_{test}$.
>
> Based on this idea, our goal is to quantify the change in test loss $L$ for $x_{test}$ resulting from adding $z_0$. To achieve this, inspired by the influence function, we define Eq.(6) to denote he parameters after training with $z_0$, which can be obtained by setting $\epsilon=\frac{1}{|D_{pre}|}$ in Eq.(6). On this basis, we further quantify the testing loss $L$ leveraging Taylor approximation as shown in Eq.(8) and (9), which serve as the foundation for subsequent theoretical analysis. Thus, in fact, Eq.(6) is a conceptual and intermediate tool for theoretical analysis rather than representing actual training on $D_{pre}$ and the demonstration $x$ simultaneously. In response to your comments, we will incorporate the above discussion and clarification to make our paper clearer.
>
> $\bf{Q3}$: In Eq.(23), $\lambda_1$ is misused. There are some typos.
>
> $\bf{A3}$: Thanks for your meticulous review and pointing out these issues! We will carefully correct the misuse of $\lambda_1$ and the typos in the revised version.

---

### Official Review · Reviewer_spYJ · 2025-03-11

**Overall Recommendation:** 4

**Summary:**

This paper aims to explore the underlying mechanism of in-context learning (ICL). To this end, the authors first theoretically analyze the influence of the demonstrations on inference performance, where they prove that the performance is bounded by an LLM-oriented semantic similarity and the demonstration stability. Then, based on this finding, they proposed a LMS3 method for demonstration selection. From extensive experiments on two widely-used LLM backbones and multiple few-shot settings, they validate the superiority of LMS3.

## update after rebuttal
The authors have addressed my concerns, and l  would like to retain my positive score.

**Claims And Evidence:**

I believe that the claims are supported by clear and convincing evidence. On one hand, the authors provide fundamental theoretical analyses to reveal the impact of demonstrations on ICL performance, offering a solid foundation for the proposed method. On the other hand, they conduct experiments with two representative LLMs and compare them against 10 baselines under various few-shot settings. Therefore, I believe the experiments are sufficient to validate the effectiveness of the proposed method.

**Essential References Not Discussed:**

I think the related work has been discussed sufficiently.

**Experimental Designs Or Analyses:**

Yes, I have carefully checked the soundness and validity of the experimental designs and analyses. Specifically, I examined the correctness of the experimental setups to ensure that they align with standard practices in the field. I also examined the selection and preprocessing of the datasets used for evaluation. Additionally, the experiments have been run multiple times to ensure the stability of the findings. No significant issues were identified, and the experimental results are consistent with theoretical expectations.

**Methods And Evaluation Criteria:**

The proposed method has strong theoretical foundation and is well designed for the problem. This paper employs answer accuracy as the evaluation metric, which is a widely used and reasonable setting. This choice ensures the reliability of the results.

**Other Comments Or Suggestions:**

I found some typos, including:

--Line 87, “remain”->”remains”

--Line 116, “satisfies”->”satisfying”

--Line 265, “suggests”->”suggest”

**Other Strengths And Weaknesses:**

Strengths:

1. The paper provides a rigorous theoretical analysis of ICL, revealing the importance of LLM-oriented semantic similarity and inference stability for reasoning performance. These theoretical findings offer a deeper understanding of when and why demonstrations help or hurt ICL.

2. The proposed LMS3 method is simple but practical and efficient. Notably, the demonstration rejection mechanism is a novel contribution. I think it is the first exploration of when ICL should not be used. This perspective fills an important gap in existing research.

3. The empirical evaluation is thorough, covering two LLM backbones, 10 baselines, and several few-shot settings. Additionally, the experiments have been run multiple times and the authors provide the confidence intervals. These results consistently support the theoretical claims and demonstrate the robustness of LMS3.

4. The paper is clearly written, well-structured, and easy to reproduce.

There are some minor issues with this paper:

1. I think the theoretical analyses are well-founded, but I hope to see if the authors could give more discussions about how to extend them to other domains or tasks (please see Question 1 below).
2. More experiments could be conducted to validate the necessity of the rejection mechanism (please see Question 2 below).
3. There exist some typos.

**Questions For Authors:**

1. The theoretical analyses in this paper are highly general. Therefore, I suggest the authors to discuss the possibility to extend them to other domains or tasks (e.g., multi-choice QA).

2. Apart from the experiments in Section 5.5., I think the authors can combine the rejection mechanism with other demonstration selection method to further validate its necessity.

3. Just for curiosity, to measure the Inference Stability of Demonstration X, could we directly test the performance of the inference LLM on X (e.g., calculate the accuracy)?

**Relation To Broader Scientific Literature:**

This paper builds on prior work in in-context learning by providing a theoretical analysis of how demonstrations influence LLM reasoning performance. Unlike prior heuristic-based or semantics-based selection methods, the proposed LMS3 is theoretically grounded, generalizable, and introduces a novel demonstration rejection mechanism. The empirical results further strengthen its contribution by demonstrating consistent improvements across multiple benchmarks and LLMs, addressing a key limitation of previous methods that lacked robustness across settings.

**Theoretical Claims:**

I have checked the proofs of all theoretical claims (i.e., Theorem 1 and Theorem 2) and verified their correctness. Additionally, I have ensured that all assumptions and derivations are logically consistent and properly justified.

---

> ### Author Rebuttal · Authors · 2025-03-30
>
> We sincerely appreciate your recognition of our theoretical analysis, the clarity and good writing of our paper, and the strong performance of our method. As for your concerns:
>
> $\bf{Q1}$: The theoretical analyses in this paper are highly general. Therefore, I suggest the authors to discuss the possibility to extend them to other domains or tasks (e.g., multi-choice QA).
>
> $\bf{A1}$: Thanks for your recognition of our theoretical analysis and this valuable suggestion. We appreciate the opportunity to discuss its broader applicability.
>
> Although the motivation of our paper stems from observations in mathematical reasoning task, the underlying principles can be applied more broadly. Indeed, as discussed in Appendix E, our conclusions can be extended to other tasks beyond those explored in this work. This is because our theoretical analyses are based on a general setup of transformer architecture and the relationship between demonstrations and test samples. As long as a task can benefit from demonstration-based prompting (e.g., multi-choice QA as you mentioned), our theoretical conclusions about LLM-oriented Semantic Similarity and Inference Stability of Demonstration in Eqs. (21) and (22) remain applicable.
>
> To validate this, we applied our method to CommonsenseQA dataset in Section 5.7, which is a large-scale commonsense benchmark and has been widely used in ICL research[1,2,3]. As shown in Table 4, our method still achieves the best performance, further demonstrating its general applicability.
>
> We sincerely appreciate your insightful comment and are very willing to explore the performance of our method on more tasks for future research. We will also enrich our discussion section in the revised version.
>
> [1] Compositional Exemplars for In-context Learning.
>
> [2] In-Context Learning with Iterative Demonstration Selection.
>
> [3] Rethinking the Role of Demonstrations: What Makes In-Context Learning Work?
>
> $\bf{Q2}$: Combine the rejection mechanism with other demonstration selection method to further validate its necessity.
>
> $\bf{A2}$: Thanks for your constructive suggestion! Following your suggestion, we supplement additional experiments using LLaMA3-8B as the backbone as follows. Specifically, we apply our demonstration rejection mechanism to all baselines, denoted as "+Our" in the table below.
> |LLaMA3-8B| MAWPS | GSM8K | MATH |
> |-|-|-|-|
> | Random|0.951|0.813|0.330|
> | +Our|0.952|0.818|0.349|
> | Best-validate|0.932|0.817|0.332|
> | +Our|0.941|0.829|0.344|
> |TF-IDF|0.945|0.803|0.344|
> | +Our|0.946|0.818|0.351|
> | BM25|0.932|0.805|0.334|
> | +Our|0.934|0.812|0.335|
> | T5|0.948|0.817|0.330|
> | +Our|0.953|0.828|0.333|
> | BGEM3|0.938|0.802|0.340|
> | +Our|0.941|0.822|0.350|
> | OpenAI|0.965|0.809|0.346|
> | +Our|0.973|0.818|0.347|
> | SPELL|0.945|0.821|0.343|
> | +Our|0.946|0.826|0.345|
> | Influence|0.929|0.800|0.333|
> | +Our|0.935|0.810|0.340|
> | IDS|0.920|0.808|0.330|
> | +Our|0.932|0.823|0.346|
>
> The results consistently show that our mechanism enhances all baselines, regardless of whether they rely on retrieval-based similarity metrics (e.g., TF-IDF, BM25) or influence-based strategies. This suggests that our rejection mechanism can serve as a general enhancement technique that improves the robustness and effectiveness of various demonstration selection approaches. This also highlight the necessity of considering when to include a demonstration in in-context learning, rather than always providing demonstrations indiscriminately. Besides, our method leads to performance improvements across different datasets, demonstrating its broad applicability.
>
> Thank you again for your valuable suggestion! We will incorporate this experiment and its analysis into the revised version to further support our findings.
>
> $\bf{Q3}$: Just for curiosity, to measure the Inference Stability of Demonstration $X$, could we directly test the performance of the inference LLM on $X$ (e.g., calculate the accuracy)?
>
> $\bf{A3}$: Thanks for your insightful question! Yes, we believe that directly testing the performance of demonstration $X$, such as calculating accuracy, can serve as a direct measure of the Inference Stability of $X$. This provides a straightforward way to assess its stability.
>
> However, one potential challenge is that achieving a reliable measurement of stability in this way might require multiple calls to the LLM. This is because a single inference may not fully reflect the model's performance over time, while averaging results from multiple runs could give a more accurate measurement of the demonstration's stability. Consequently, this method could incur additional computational costs due to the need for repeated evaluations of the demonstration.
>
> $\bf{Q4}$: I found some typos.
>
> $\bf{A4}$: Thanks for your meticulous review and pointing out these typos! We will carefully correct them in the revised version.

---

### Official Review · Reviewer_2APu · 2025-03-14

**Overall Recommendation:** 4

**Summary:**

In-context learning has been a key driver of LLM performance over the past few years. However, the performance of a model can vary (and sometimes even be negatively impacted) based on the content of the few-shot demonstrations provided in-context. This work provides a theoretical analysis of the conditions under which ICL is beneficial and find that performance depends on 2 keys factors :1) The semantic similarity of the demonstration to the test problem and 2) The inference stability of the demonstration which indicates how easily the LLM can solve the demonstration itself. Building on their theoretical insights, the authors present LMS3, a simple algorithm to select the demonstration(s) for a given problem by trading-off between the 2 objectives defined above. Results show that their proposed algorithm improves performance over existing ICL selection methods and surprisingly, they also find that for some questions, having no demonstrations is actually beneficial!

**Claims And Evidence:**

Yes, theoretical claims are supported by appropriate proofs - however I have not checked the math carefully. Their proposed algorithm is validated on 3 standard benchmarks for mathematical reasoning and shows strong performance.

**Essential References Not Discussed:**

N/A

**Experimental Designs Or Analyses:**

Yes, experimental design and analysis is valid.

**Methods And Evaluation Criteria:**

Yes, standard evaluation criteria (accuracy) are chosen and appropriate benchmarks are selected. Confidence intervals are also provided for the main set of results.

**Other Comments Or Suggestions:**

N/A

**Other Strengths And Weaknesses:**

- This a very-written paper with theoretical analysis backed by a practical algorithm. The paper also does a good job of building intuition and the final findings align well with prior work in the area.
- The algorithm presented requires white box access to the LLM. However, given the strong generalization performance across LLMs, this might not be an issue.

**Questions For Authors:**

- In the K-shot setting, is it fair to treat the demonstrations as independent of each other? Aren't there potentially significant interaction effects that need to be accounted for?
-  Can insights from this work potentially be used to design a method that generates effective demonstrations? That is, use some form of optimization to generate a X that maximizes the score function. This might result in better performance compared to a fixed offline demonstration set.
- By how much would the inference runtime for a query be affected (compared to random demonstration) if LMS3 is used using a fixed demonstration dataset. If I understand currently, since the embeddings of all demonstrations are computed independently and stored, the overhead should be minimal?

**Relation To Broader Scientific Literature:**

This paper provides theoretical grounding to empirical findings that have been observed in prior work.

**Theoretical Claims:**

I did not check the correctness of proofs.

---

> ### Author Rebuttal · Authors · 2025-03-30
>
> We sincerely appreciate your affirmation of the effectiveness of our experiments, the good writing of our paper, and the significance of our work. As for your concerns:
>
> $\bf{Q1}$: The algorithm presented requires white box access to the LLM. However, given the strong generalization performance across LLMs, this might not be an issue.
>
> $\bf{A1}$: Thanks for your valuable comment! Yes, as our theoretical analysis suggests, the influence of demonstrations is determined by two key factors: LLM-oriented Semantic Similarity and Inference Stability of Demonstration. As defined in Eqs. (21) and (22), both factors are computed based on model representations. While they require white-box access to the model, this aligns with intuition, as different LLMs may benefit from different demonstrations depending on their own capabilities and characteristics.
>
> To address the concern about generalization, as you pointed out, we examined this aspect in Section 5.6 by applying the demonstrations selected by our LMS3 method using Llama3-8B as backbone to ChatGPT and GPT-4. As shown in Table 3, these demonstrations still significantly improve their accuracy, demonstrating the strong generalization capability of our method. Under these conditions, LMS3 continues to achieve the best overall performance, highlighting its potential to provide valuable demonstrations even when working with closed-source LLMs in practical applications.
>
> We sincerely appreciate your thoughtful comment and hope this explanation addresses your concerns.
>
> $\bf{Q2}$: In the K-shot setting, is it fair to treat the demonstrations as independent of each other?
>
> $\bf{A2}$: Thank you for the insightful question! In this work, our theoretical analysis considers the most fundamental case and starts with a single attention layer in transformers. Under this setup, as shown in Eq. (17), the influences of different demonstrations on the representation of the test sample $h_{test}$ follow an almost linear relationship, which allows us to treat them independently. In a full transformer architecture where deeper interactions occur (e.g., multi layers of cross-attention), the representations of different demonstrations will interact in more intricate ways, which may lead to complex effects on $h_{test}$ that could not be directly measurable. Therefore, our findings provide a foundational understanding that can offer insights into practical scenarios.
>
> Moreover, even under this theoretical simplification, our method LMS3 consistently outperforms the baselines across all settings from 2-shot to 4-shot (Figure 3). This validates the feasibility of our theoretical results and the effectiveness of our method. Following your comment, we are very willing to further explore the impact of different demonstration combinations in the future.
>
> $\bf{Q3}$: Can insights from this work potentially be used to design a method that generates effective demonstrations? This might result in better performance compared to a fixed offline demonstration set.
>
> $\bf{A3}$: Thanks for your constructive suggestion! We fully agree the idea that our work can be used for effective demonstration generation. This is because our theoretical analysis sheds light on what characteristics contribute to their effectiveness and we can easily estimate the scores of demonstrations by Eq.(24) in our paper. This presents an exciting direction worth further exploration.
>
> Further considering this idea, we think the only challenge in implementation is ensuring that the generated demonstrations have correct answers. While a fixed offline demonstration set allows for manual curation to guarantee correctness, dynamically generated demonstrations require additional mechanisms to verify their validity and reliability. Developing such mechanisms remains an open and important question.
>
> We greatly appreciate your thought-provoking idea and will supplement the discussion it in our revised version.
>
> $\bf{Q4}$: Inference runtime for a query if LMS3 is used using a fixed demonstration dataset. If I understand currently, the overhead should be minimal?
>
> $\bf{A4}$: Thanks for your valuable question. Yes, your understanding is correct. Since the representations of all demonstrations can be precomputed in advance, the inference process in our LMS3 only involves encoding the test sample and retrieving the relevant precomputed information. This ensures that our method achieves the minimal computational complexity during inference as shown in Table 1.
>
> We appreciate your thoughtful question and hope this clarifies our approach.

---

> > ### Comment · Reviewer_2APu · 2025-04-02
> >
> > I thank the authors for taking the time to answer my questions! This paper presents interesting theoretical analysis and backs it up with interesting empirical analysis. The few limitations (generalizability and the assumption that demonstrations are independent of each other) will be interesting follow-up works.
> >
> > I have raised my score.

---

> > > ### Author Response · Authors · 2025-04-02
> > >
> > > We sincerely appreciate your constructive comments and valuable feedback! We will also incorporate our discussion on these aspects into the revised version. Thank you again for your time and for raising your score!

---

### Decision · Program_Chairs · 2025-05-01

**Decision:**

Accept (poster)

**Comment:**

This paper investigates the effectiveness of in-context learning from a theoretical perspective and derives a lower-bound (Theorem 1) on the prediction error over the test example by its similarity to the in-context samples and the model stability to these examples. Using Theorem 1, the paper derives a scoring scheme, LMS3, on few-shot examples that could potentially be useful for improving the accuracy on the test examples. Experiments are provided using two LLMs on varied reasoning tasks, demonstrating promising results.

The paper is organized well and received overall positive reviews. All the reviewers appreciated the theoretical insights and the empirical results. There are minor concerns on the generalizability of the approach and inference time of LMS3 (2APu), extension of the insights to other domains beyond mathematical reasoning (spYJ), and extension of the LMS3 scheme to other demonstration schemes (spYJ).  Authors addressed these concerns and provided additional results demonstrating the generlizability of the scheme to an elaborate set of demonstration selection methods with promise.

There are three key outstanding concerns in the paper, namely:
1) the proposed approach needs a scheme to derive the stability of the in-context examples, which would ideally need access to the network weights,
2) simplified analysis of the approach to a single layer transformer setting with linearized softmax.
3) the applicability/empirical analysis of the approach only to mathematical reasoning problems,

AC thinks these issues may not be critical given the theoretical focus of the paper.  That being said, AC makes a note of the concern raised by Reviewer voNS regarding the derivations in Theorem 1 that uses the annotations in (6)-(ii). Specifically, the theoretical approach in the paper is to investigate in-context learning through the lens of few-shot training and (6) considers a setting using hypothetical weights \hat{W}_\eps, z0 that could have been obtained, had the in-context examples been part of training (as few-shot examples). While the idea simplifies the theoretical derivations, the subsequent derivations may not necessarily be truthful to understanding in-context examples as standard in-context formulations do not assume any updates to the network weights and are usually test-time prompt samples. Thus, the paper may be improved by providing more clarity into these underlying assumptions.

All the reviewers recommend acceptance of this paper. AC concords with the reviewers' sentiment and see that the paper has value in terms of theoretical insights (that could be explored further) as well as elaborate empirical results. As such AC recommends acceptance. Authors are recommended to address reviewers' comments, especially on clarifications to (6), in the camera-ready.